# The Phosphoglycerate Kinase (PGK) Gene Family of Maize (*Zea mays* var. B73)

**DOI:** 10.3390/plants9121639

**Published:** 2020-11-24

**Authors:** Julio A. Massange-Sánchez, Luz E. Casados-Vázquez, Sheila Juarez-Colunga, Ruairidh J. H. Sawers, Axel Tiessen

**Affiliations:** 1Departamento de Ingeniería Genética, CINVESTAV Unidad Irapuato, Irapuato 36821, Mexico; edith.casados@ugto.mx (L.E.C.-V.); sheila.juarez@cinvestav.mx (S.J.-C.); axel.tiessen@cinvestav.mx (A.T.); 2Unidad de Biotecnología Vegetal, Centro de Investigación y Asistencia en Tecnología y Diseño del Estado de Jalisco A.C. (CIATEJ) Subsede Zapopan, Guadalajara 44270, Mexico; 3Life Science Division Food Department, University of Guanajuato Campus Irapuato-Salamanca, Irapuato, Guanajuato 36500, Mexico; 4Department of Plant Science, The Pennsylvania State University, State College, PA 16801, USA; rjs6686@psu.edu; 5Laboratorio Nacional PlanTECC, Ciudad de México C.P. 06020, Mexico

**Keywords:** phosphoglycerate kinase, *Zea mays*, gene family, gene expression, phylogeny

## Abstract

Phosphoglycerate kinase (PGK, E.C. 2.7.2.3) interconverts ADP + 1,3-bisphospho-glycerate (1,3-bPGA) to ATP + 3-phosphoglycerate (3PGA). While most bacteria have a single *pgk* gene and mammals possess two copies, plant genomes contain three or more *PGK* genes. In this study, we identified five *Pgk* genes in the *Zea mays* var. B73 genome, predicted to encode proteins targeted to different subcellular compartments: *ZmPgk1*, *ZmPgk2,* and *ZmPgk4* (chloroplast), *ZmPgk3* (cytosol), and *ZmPgk5* (nucleus). The expression of *ZmPgk3* was highest in non-photosynthetic tissues (roots and cobs), where PGK activity was also greatest, consistent with a function in glycolysis. Green tissues (leaf blade and husk leaf) showed intermediate levels of PGK activity, and predominantly expressed *ZmPgk1* and *ZmPgk2*, suggesting involvement in photosynthetic metabolism. *ZmPgk5* was weakly expressed and *ZmPgk4* was not detected in any tissue. Phylogenetic analysis showed that the photosynthetic and glycolytic isozymes of plants clustered together, but were distinct from PGKs of animals, fungi, protozoa, and bacteria, indicating that photosynthetic and glycolytic isozymes of plants diversified after the divergence of the plant lineage from other groups. These results show the distinct role of each PGK in maize and provide the basis for future studies into the regulation and function of this key enzyme.

## 1. Introduction

Phosphoglycerate kinase (PGK, E.C. 2.7.2.3) catalyzes the reversible conversion of ADP + 1,3-bisphosphoglycerate (1,3-bPGA) to ATP + 3-phosphoglycerate (3PGA). It is a key enzyme involved in both glycolysis and photosynthesis (Figure 1A). In the glycolytic pathway of all organisms, PGK produces ATP and recovers the energy investment of the hexokinase (HK) and phosphofructokinase (PFK) enzymes. In photosynthetic organisms, PGK catalyzes a key step in the Calvin–Benson–Bassham (CBB) cycle, consuming ATP from the light reactions and leading to the reduction of 3-PGA to sugar-phosphates [1]. It is a readily reversible reaction, in which substrates and products can be either delivered or withdrawn [2]. The sequential action of PGK and glycerol-aldehyde-3P-deshydrogenase (GAPDH) is the only way by which all the carbons from 3PGA, produced by Rubisco, can be stoichiometrically re-converted into Ribulose-1,5-bP, with some gain of triose phosphate molecules (Figure 1A).

PGK functions as a monomeric enzyme, with a molecular mass of ~45 kDa [3], which is about average for eukaryotic proteins [4,5]. The tertiary structure is organized into two separate domains: The N-terminal domain, which binds either 3PGA or 1,3-bPGA, and the C-terminal domain, which binds to either ATP or ADP [3]. PGK is a typical hinge-bending enzyme that shows a substrate-assisted domain closure mechanism [6,7] similar to other metabolic kinases, such as HK and PFK [8]. Each domain is composed of a central β-sheet, sandwiched by two α-helical layers (Rossmann fold), which are hinged together by a flexible helix that permits them to close for catalysis [9,10,11].

Most bacterial genomes contain a single gene-encoding PGK (Figure 1B) [12,13]. In eukaryotes, the ancestral PGK appears to have been lost from the mitochondrial genome and anaerobic glycolysis is performed in the cytosol. In *Homo sapiens*, cytosolically localized PGK is encoded by two nuclear genes, *HsPGK1* and *HsPGK2*, which are differentially regulated in various tissues [14,15]. In plants, there are cytosolic and chloroplastic PGK isozymes, both encoded by nuclear genes [16,17,18,19]. It has been hypothesized that plant *PGK* genes were acquired from a cyanobacterial ancestor by horizontal gene transfer [1] such that the chloroplastic isozyme would be homologous to cyanobacterial and algal proteins, whereas the plant cytosolic isozyme would be homologous to glycolytic enzymes in other eukaryotes.

Maize (*Zea mays*) is one of the main sources of nutrition for humans and domesticated animals, with an average annual world production at ≈1100 million tons in the last five years (http://www.fao.org/faostat/en/#data). However, it is necessary to increase maize crop production to feed the whole population in the coming decades. The growth stages of maize are divided into vegetative stages (V0-V14) and reproductive stages (R1-R6). In reproductive stage 1 (R1) silks emerge; pollen is captured by silks and germinates within the next 24 hours with the consequent fertilization. From R1 until reproductive stage 3 (R3) different phenomena occur in the grain. For example, the cell division and cell expansion are activated to increase the potential of grain to accumulate sugars and starch [20]. Further, in terms of looking alternatives to increase yield and crop productivity, it is necessary to improve the efficiency of the whole plant mechanism, which are mainly related to address the C-resources in glycolysis and/or photosynthesis, such the cases of PGK proteins. In this way it will ensure a good yield and better quality in different tissues.

In this study, we characterized the PGK enzyme family of maize. We found that the *Zea mays* var. B73 genome contains five genes putatively encoding PGK (*ZmPgk1-ZmPgk5*), although *ZmPgk4* appears to be non-functional. To clarify the distinct role of each PGK, we conducted a molecular and a biochemical characterization of the maize isozymes, coupled with the measurement of gene expression and PGK activity in different tissues. Phylogenetic analysis distinguished nuclear, cytosolic and chloroplastic isozymes in plants from other groups, and did not support the hypothesis that *PGK* genes were acquired by horizontal transfer from a cyanobacterial ancestor.

## 2. Materials and Methods

### 2.1. Identification of PGK-Encoding Genes in the Maize Genome

The B73 genome AGP_v4 of the MaizeSequence database (www.maizesequence.org) was searched to identify maize PGKs. We first search using the keyword “PGK”, then a protein BLAST was made with a functionally validated PGK sequence reported in GenBank for humans (ACG36709.1) or *E. coli* (AAC75963.1). A HMMER version 3.1b3 (http://hmmer.org/) [21] search was run against AGP_v4 maize genome assemblies using both N-terminal (IPR015824) and C-terminal (IPR015901) PGK domains, extracted from the Pfam database release 30 [22]. The list was manually curated, considering the total length of the proteins, ORF (open reading frame), and local similarities. The final maize PGK set was consistent with CornCyc-predicted PGKs (https://corncyc-b73-v4.maizegdb.org/CORN/NEW-IMAGE?type=NIL&object=EC-2.7.2.3&redirect=T).

### 2.2. Bioinformatic Analysis

Subcellular targeting was predicted with Predotar [23] (https://omictools.com/predotar-tool), Wolf PSORT [24] (https://wolfpsort.hgc.jp/), ChloroP [25], and TargetP [26]; the last two software tools are available in ExPASy Bioinformatics Resource Portal (https://www.expasy.org/).

Molecular weight (MW) and isoelectric point (pI) were calculated with and without predicted signal peptide sequences using ProtParam [27] (https://www.expasy.org/). EMBL (Lehninger 1979) (https://www.ebi.ac.uk/) and Sequence Manipulation Suite (SMS) [28] (http://www.bioinformatics.org/sms2/) tools were also used to calculate the isoelectric point (pI). Possible N-glycosylation sites were predicted with NetNGlyc 1.0 server (http://www.cbs.dtu.dk/index.html). InterProScan [29] (https://www.ebi.ac.uk/) was used for domain searching, and the illustration was constructed with DOG 2.0 [30] (http://dog.biocuckoo.org/).

### 2.3. Modeling of PGK Protein Structure

The five coding sequences of ZmPGK1 to ZmPGK5 obtained from www.maizesequence.org were submitted to the I-TASSER website (http://zhanglab.ccmb.med.umich.edu/I-TASSER/) [31,32] to generate five ZmPGK in silico structural models. The protein sequences for ZmPGK1, ZmPGK2, and ZmPGK4 were processed (i.e., transit peptide was removed) before I-TASSER analysis. Molecular graphics were generated and structural analyses of the protein models were performed with UCSF Chimera v1.6.1 [33].

### 2.4. Promoter Analysis

One kb of sequence upstream of the ATG was obtained for maize and *Arabidopsis PGK* genes from Ensembl Plants genome annotation [34] (https://plants.ensembl.org/index.html), last accessed 03 September 2018) and compared with the PLACE database of plant cis-acting regulatory DNA motifs [35] (https://sogo.dna.affrc.go.jp/cgi-bin/sogo.cgi?lang=en&pj=640&action=page&page=newplace). Cis-elements were ordered in the heat map according to a hierarchical clustering of the number of motifs present in the promoters using a Pearson correlation distance function [36]. Only motifs present in more than one sequence were included in the analysis.

### 2.5. Phylogenetic Analysis

Amino acid sequences of the five PGKs in maize genome B73 AGP_v4 and fungi, Protista, bacteria, animal, and plant PGKs (Appendix A) were retrieved from GenBank [37] (https://www.ncbi.nlm.nih.gov/), and Ensemble public databases [34] (https://plants.ensembl.org/index.html). Complete sequences were aligned using MUSCLE [38] under default parameters (Appendix A). Maximum likelihood analysis of 99 sequences was performed with PhyML [39] using the WAG model and a gamma distribution. The reliability of branches was estimated by bootstrapping 1000 replicates using the same gamma parameters in MEGA 7.0 [40]. Phylogenetic trees were constructed using the neighbor-joining method with the Poisson model, uniform rate, and pairwise deletion parameters, results were compared using maximum likelihood clustering.

### 2.6. RNA Isolation and Reverse Transcription

Total RNA was isolated at reproductive stages R1 and R3, from nine different maize tissues: Leaf blade (LB), leaf sheath (LS), node (ND), internode (IN), husk leaf (HL), peduncle (PD), corncob (CC), grain (GR), and root (RT). Trizol extraction was performed in accordance with the manufacturer’s instructions (Invitrogen). Total RNA was precipitated with 2.5 M LiCl, and the RNA pellet was re-suspended in sterilized water to a final volume of 50 µL. RNA integrity was assessed by electrophoresis on 1% agarose gel. Total RNA (1 µg) was reverse transcribed with 100 U of SuperScript II RT (SSII) (Invitrogen) using 25 µg/µL oligo dT primer, in accordance with the manufacturer’s instructions.

### 2.7. Quantitative Real-Time Reverse Transcription Polymerase Chain Reaction (qRT-PCR)

qRT-PCR was performed with the StepOnePlus™ Real-Time PCR System (Applied Biosystems) using SYBR Green JumpStartTM TaqReadyMixTM (Sigma) and a final concentration of 10 TrisHCl, pH8.3; 50 KCl; 3.5 MgCl_2_; 0.2 mM each of dNTP (dATP, dCTP, dGTP, dTTP); 0.05 U/µL Taq DNA polymerase; JumpStart Taq antibody; 2X internal reference dye and SYBR Green I. Mastermix, made using 20 ng/µL reverse transcribed total RNA; 0.3 µM appropriate oligonucleotides; and SYBR Green JumpStartTM TaqReadyMixTM, 2X. The PCR conditions were denaturation (95 °C for 10 min); amplification and quantification 40× (95 °C for 15 s, 60 °C for 1 min); melting (60–95 °C, with a heating rate of 0.3 °C per second). Data analysis was conducted by importing data from the CFX Manager to Excel using an absolute expression method for calculations [41,42]. The elongation factor EF1α (Zm00001d036904) was used as a reference gene [43]. Primers (Appendix A) were designed for gene specificity and optimized for amplicon size = 100–300 bp, GC content = 40–60%, and annealing temperature = 65 ± 3 °C. The exon–intron structure of genes was used to discriminate amplification of genomic DNA from transcribed cDNA.

### 2.8. Enzyme Activity Measurements

Maize plants (inbred line B73) were grown in the greenhouse under environmental conditions in Irapuato, Mexico. For assaying PGK activity, five maize tissues (LB, HL, CC, RT, and ST; stem) were rapidly harvested and frozen in liquid nitrogen. Approximately 500 mg of fresh tissue was used for each assay. The frozen tissue was finely ground into a powder, using a metal mill (Retsch, Germany), and extracted with buffer A (50 mM Tris HCl, pH 8). The slurry was centrifuged at 10,000 g for 20 min at 4 °C. The supernatant was immediately used to evaluate PGK activity by measuring the consumption of NADH (CN 10128058001, JJJR Cientifica (www.jrcientifica.com)), at 340 nm, in a coupled reaction with glyceraldehyde-3-phosphate dehydrogenase (GAPDH) (CN G2267, Sigma (https://www.sigmaaldrich.com/mexico.html)). The standard 100 µL mixture contained 5 mM MgSO_4_, 2 mM ATP (CN 10127523001, JJJR Cientifica (www.jrcientifica.com)), 2 mM 3PGA (Sigma USA), 2 mM NADH, and 0.1 mg/mL GAPDH buffered in 100 mM Tris-HCl (Karal, Mexico), pH 7.5. Reactions were initiated by adding 10 µL of protein extracts at 25 °C and followed a microplate reader for at least 20 min. Control assays were performed by omitting individual substrates. Soluble protein was measured by adding Coomassie Brilliant Blue dye (Sigma, Madison, WI, USA) [44] with bovine serum albumin (BSA) (CN 05470, Sigma (https://www.sigmaaldrich.com/mexico.html)) as standard. To estimate substrate affinity to 3PGA and/or ATP for each tissue type, Michaelis Menten kinetic was performed varying concentration of 3PGA or ATP.

## 3. Results

### 3.1. Identification of PGK Isozymes in Maize Genome (ZmPGKs)

PGK is essential in the metabolism of most living organisms, and their sequence has remained highly conserved throughout evolution. To identify maize *Pgk* genes, we searched the maize genome (B73v4) using bacterial and human PGKs. A total of 117 maize gene-models were identified to show weak similarity to PGK proteins. After manually curation, a final set of five putative PGK-encoding genes was selected (*ZmPgk1*: Zm00001d038579_T001.1, *ZmPgk2*:Zm00001d010672_T001.1, *ZmPgk3*: Zm00001d015376_T002.1, *ZmPgk4*: Zm00001d043194_T001.1, and *ZmPgk5*: Zm00001d032867_T002.1; Table 1). The final gene set was consistent with that identified by the CornCyc metabolic pathway database (https://corncyc-b73-v4.maizegdb.org/CORN/NEW-IMAGE?type=NIL&object=EC-2.7.2.3&redirect=T). *ZmPgk1* was located on chromosome (chr) 6, *ZmPgk2* on chr 8, *ZmPgk3* on chr 5, *ZmPgk4* on chr 3, and *ZmPgk5* on chr 1 (Appendix A). At the protein level, ZmPGK1 and ZmPGK2 shared the highest identity (95%) between all ZmPGKs (Table 1). ZmPGK3 and ZmPGK4 shared over 80 and 66% identity with ZmPGK1, respectively. While ZmPGK5 was the protein with the least identity to other PGKs; ZmPGK1 and ZmPGK5 showed 22% of identity.

### 3.2. Analysis of the ZmPGK Protein Sequences

Both animal and plant PGKs contain nine conserved motifs (M1-M9), distributed between N-terminal (M1-M4) and C-terminal (M5-M9) domains (Figure 2). ZmPGK1–ZmPGK3 were predicted to contain all nine motifs, ZmPGK4 only two motifs, and ZmPGK5 five motifs (Figure 2). ZmPGK5 contained an additional methyltransferase domain (IPR003358) in the C-terminus (Figure 2). The sequences of the ZmPGK proteins were analyzed to predict size, subcellular location, and isoelectric point (pI) (Table 2). The molecular weights were predicted to be ~42 kDa for the mature ZmPGK1–ZmPGK3 proteins. ZmPGK4 (NP_001167912.1) was smaller (~32.9 kDa) [45], whereas ZmPGK5 was larger (~56.4 kDa) (Table 2). ZmPGK1–ZmPGK4 had a similar pI of ~5, while ZmPGK5 had a pI of 6. ZmPGK1, ZmPGK2, and ZmPGK4 included chloroplast-targeting signals of 71, 68, and 53 residues, respectively (Table 2). ZmPGK3 lacked a transit peptide. ZmPGK5 contained a predicted nuclear localization signal. ZmPGKs were predicted to contain glycosylation sites, except for ZmPGK3 (Table 2). Based on predicted subcellular localization and sequence analysis, we predicted ZmPGK1 and ZmPGK2 to be photosynthetic PGKs and ZmPGK3 to be involved in glycolysis. ZmPGK4 does not appear to encode a functional protein, while the biological role of ZmPGK5 is not clear.

### 3.3. In Silico Structural ZmPGK Models and Possible Interactions with Their Substrates

To identify residues of the ZmPGKs that might interact with different substrates, a series of in silico structural models were generated (Figure 3A). To date, no structures have been determined for plant PGKs. In the absence of an available structure for a plant PGK, the maize PGKs were modeled based on structural scaffolds from bacterial and animal sources [11,46,47,48,49]. We identified seven predominantly basic residues in the N-terminal domain (3PGA binding) of ZmPGK1 that were predicted to play an important role in either selectivity, binding, or catalysis: D99, N101, R115, H138, R141, R196, and R227 (Figure 3B). The residues H138, R141, R136, and R227 were predicted to interact with the phosphate group of 3PGA (Figure 3B). The residues R115, D99, and N101 were predicted to interact with the carboxyl group of 3PGA (Figure 3B). In the C-terminal (nucleotide binding) domain, we identified four important residues: D433, S434, N395, and R276 with the potential to stabilize the nucleotide diphosphate (Figure 3C); E402 predicted to interact with the adenine in ATP (Figure 3C).

### 3.4. Differential Expression of ZmPgk Genes

To investigate the tissue expression patterns of *ZmPgk* genes, we analyzed two reproductive growth stages of maize plants cultivated under greenhouse conditions. We performed absolute quantification analysis by calculating PCR efficiency in each reaction [41,42] of all *ZmPgk* genes. We analyzed leaf blade (LB), leaf sheath (LS), node (ND), internode (IN), husk leaf (HL), peduncle (PD), corncob (CC), grain (GR), and root (RT) samples from plants in two developmental stages, R1 and R3 (Figure 4). Comparison of the absolute expression level between *ZmPgk* genes and *ZmEf1α* reference gene is shown in (Appendix A). In the R1 stage, *ZmPgk3* was the highest expressed in all organs except for LB (Figure 4C), with predominant expression in non-green tissues (Figure 4C). *ZmPgk1* and *ZmPgk2* were more highly expressed in green tissues (LB, LS, ND, IN, HL, and PD) than non-green tissues (CC, GR, and RT), with *ZmPgk1* consistently expressed about ten-fold higher than *ZmPgk2* (Figure 4A,B). In contrast to the R1 stage, during the R3 stage *ZmPgk1* showed higher expression (LB and HL) than *ZmPgk3* (Figure 4E). During the R3 stage, *ZmPgk3* was expressed in LB, ND, and PD, but not other tissues (Figure 4G). *ZmPgk5* was weakly expressed (Figure 4D,H) and *ZmPgk4* was not detectable in any of the analyzed tissues and stages. We also inspected the expression profile of the *ZmPgk* genes in publicly available maize RNA-seq data sets (http://bar.utoronto.ca/efp_maize/cgi-bin/efpWeb.cgi) [50,51,52]. A consistent pattern was observed in which *ZmPgk3* was the highest expressed, *ZmPgk1* and *ZmPgk2* showed similar expression, *ZmPgk5* was the lowest expressed, and *ZmPgk4* was not detected (Figure 4).

### 3.5. Identification of Cis-Regulatory Elements Correlated with Tissue-Specific Expression of ZmPGKs

To identify candidate cis-elements driving the tissue-specific expression of the *ZmPgk* genes, we compared the co-occurrence of specific promoter motifs in both maize and *Arabidopsis*. In total, 96 distinct cis-regulatory elements were found in all five ZmPGK and three AtPGK promoter sequences (Appendix A). Maize and *Arabidopsis* PGK promoter sequences share 50% of cis-elements (48 different motifs), whilst 43% (41 different motifs) and 7% (7 different motifs) of cis-elements were exclusively from maize and *Arabidopsis*, respectively (Appendix A). We performed a clustering analysis of the 50 most represented cis-elements from ZmPGK and AtPGK promoter sequences (Figure 5). Among these, ASF1MOTIFCAMV, IBOXCORE, TAAAGSTKST1, SORLIP1AT, CIACADIANLELHC, GT1CONSENSUS, SORLIP2AT, and INRNTPSADB elements are linked to light regulation (Figure 5; Appendix A). CACTFTPPCA1, GATABOX, -10PEHVPSBD, PRECONSCRHSP70A, and BOXIINTPATPB elements are linked to regulation of chloroplast/plastid genes (Figure 5; Appendix A). DOFCOREZM, EBOXBNNAPA, SEF4MOTIFGM7S, GCN4OSGLUB1, and ACGTCBOX elements are linked to endosperm specific expression during seed development (Figure 5; Appendix A). Other elements, including ANAERO2CONSENSUS, MYCCONSENSUSAT, LTRECOREATCOR15, and CAATBOX1, are linked to transcriptional responses to pathogens, wounding, drought, cold, heat, copper, and sulfur (Figure 5; Appendix A).

### 3.6. Kinetic Analysis in Maize

To measure PGK activity, we sampled the five tissues with the highest expression of the cytosolic *ZmPgk3* (corncob = CC and root = RT) and plastid *ZmPgk1* and *ZmPgk2* (leaf blade = LB, husk leaf = HL, and internode and node = stem = ST). At the R1 stage, the highest PGK activity was found in the heterotrophic tissues RT, CC, and ST (Figure 6), whereas, the photosynthetic tissues LB and HL had intermediate activities (Figure 6). In addition, we estimated the substrate affinity to 3PGA and/or ATP for each tissue type (Appendix A). Substrate affinity for 3PGA was similar among most tissues (~0.5 mM), except in LB, which had a Km of ~0.84 mM (Table 3). Substrate affinity for ATP was more markedly different among tissues (Table 3): For example, in LB a Km of 0.06 mM contrasted with a Km of 0.24 mM in CC (Table 3).

### 3.7. Phylogenetic Relationships of Plant PGK Isozymes

To compare maize PGKs to other organisms, we identified further PGKs from GenBank (https://www.ncbi.nlm.nih.gov/genbank/) and phytozome (https://phytozome.jgi.doe.gov/pz/portal.html) databases. In addition, we searched the genomes of *Oriza sativa*, *Populus trichocarpa*, *Sorghum bicolor,* and *Vitis vinifera* (Table 4). The protein list was enriched iteratively and curated manually, identifying 99 PGKs from different species: 3 from archaea, 11 from bacteria, 8 from protozoa, 11 from fungi, 7 from animals, 51 from flowering plants, and 8 from green algae (Figure 7 and Appendix A). To identify key amino acids, the protein sequences of all PGKs were aligned and compared (Appendix A). Moreover, we generated a phylogenetic tree of the 99 PGKs. We included the cytosolic and glycosomal isozymes of *Trypanosoma sp*. and *Crithidia sp*. and the two cytosolic isozymes of humans in the phylogenetic analysis. The tree allowed us to identify clusters that clarified the chloroplastic, cytosolic, and nuclear classifications (Figure 7 and Appendix A). In general, PGKs were clustering according to the kingdom and the species they came from, except for nuclear plant PGKs that created a separate group (Figure 7 and Appendix A). Chloroplastic and cytosolic isozymes from plants were more related to each other than to cytosolic isozymes in animals and cyanobacteria (Figure 7 and Appendix A). Some algae contained an ancient glycolytic PGK isozyme that appears to have been lost in modern plants (Figure 7 and Appendix A). We also constructed a phylogenetic tree using IQ-TREE [53]. Similar to MEGA maximum likelihood analysis, PGKs were clustered according to the kingdom and the species they came from (Appendix A).

## 4. Discussion

Phosphoglycerate kinase (PGK) can either produce or consume ATP and 3PGA, depending on the energy status and the metabolic conditions prevailing in different subcellular compartments (Figure 1A). PGKs have been predominantly characterized in bacteria [13,46], animal [9], and protist species [54,55], and recently in *Arabidopsis* [56]. We conducted a detailed characterization to clarify the distinct role of each PGK in maize and establish the basis for functional studies of this key enzyme. The R1 and R3 stages of maize were chosen as particular phases for studying the pattern expression and enzyme activity of ZmPGKs because they are very different in their requirement of energy and metabolism. It means that in less than 20 days cellular events that define if grains can achieve their maximum potential for grain filling must occur [20]. The plant PGK isozymes involved in glycolytic and fatty acid metabolism have been investigated in castor oil seeds [57] and sunflower seeds [19]. We analyzed putative PGKs from many plant genomes in order to gain a deeper understanding of the functional differences between the glycolytic and photosynthetic PGKs from plant cells.

Plastid-localized ZmPGK1 and ZmPGK2 are predicted to be involved in photosynthetic metabolism.

Photosynthetic activity is clearly dependent on chlorophyll in leaves and other plants tissues [58]. Three PGK-encoding genes have been described in *Arabidopsis*; At3g12780 (*AtPGK1*), At1g56190 (*AtPGK2*), and At1g79550 (*AtPGK3*) (https://www.plantcyc.org/). AtPGK1 is confined exclusively in the chloroplasts of photosynthetic tissues, *AtPGK2* is expressed in the chloroplast/plastid of photosynthetic and non-photosynthetic cells, and AtPGK3 is located in the cytosol [56]. In maize, we found three PGKs (ZmPGK1, ZmPGK2, and ZmPGK4) predicted to be targeted to the chloroplast. Differences in gene expression (Figure 4) and kinetic properties (Table 3) between ZmPGK1 and ZmPGK2 might favor photosynthesis in specialized cell types, whilst ZmPGK4 looks to not be functional (see below).

In C4 plants, such as maize, the photosynthetic apparatus is partitioned between mesophyll and bundle sheath cells, which differ in their morphology and function [50]. Mutation of *AtPGK2* (At1g56190) is lethal in *Arabidopsis* [56], suggesting that the two plastid isozymes are not functionally redundant, and they play different roles in plant metabolism. Notably, the plastid *ZmPgk1* and *ZmPgk2* were more highly expressed in green than non-green tissues (CC, GR, and RT), with *ZmPgk1* consistently expressed about ten-fold higher than *ZmPgk2* both in maize R1 and in R3 growth stages (Figure 4). Proteomic surveys in fractionated maize leaves had shown that the protein ZmPGK1 (GRMZM2G089136_P01) was enriched in mesophyll cells, whereas ZmPGK2 (GRMZM2G083016_P01) was enriched in bundle sheath cells (http://ppdb.tc.cornell.edu/dbsearch/gene.aspx?id=972684) [59]. This suggests that both plastid ZmPGKs are involved in photosynthetic metabolism (ATP consumption), with different roles in maize tissues, and ZmPGK1 plays the main function in photosynthetic metabolism to optimize the growth of maize plants.

### 4.1. Cytosolic ZmPGK3 is Predicted to be Involved in Glycolysis

The PGK reaction is an essential step of the glycolytic pathway of all organisms [1]. Rosa–Téllez et al. (2018) showed that AtPGK3 accumulates specifically in the cytosol and that the gene was expressed ubiquitously in all tissues (stems, roots, siliques, leaves, and flowers). Bioinformatics analysis predicted ZmPGK3 in the cytosol and it was more highly expressed in non-green tissues (CC, GR, and RT) than in green tissues (LB, LS, ND, IN, HL, and PD) (Figure 4C,G). In addition, CC, GR, and RT are heterotrophic tissues where the highest PGK activity was found (Figure 6). Remarkably, it has been shown in *Arabidopsis* that the knockout mutation of the cytosolic PGK (*AtPGK3*) is not lethal; in fact *Atpgk3* mutant plants only show minor phenotypic changes [56]. Bacterial, fungal, and animal cells would not survive without the ATP produced by the PGK reaction, while plant cells may produce enough 3PGA and ATP from other sources, such as the non-phosphorylating GAPDH reaction or photosynthesis. ZmPGK3 contains all the motifs required to actively function in the cytosol (Figure 4), indicating its involvement in the glycolytic pathway (ATP production). Most likely a *Zmpgk3* mutant plant would not be lethal, but may well impact grain filling and cob development.

### 4.2. ZmPGK4 is a Pseudogene

The CornCyc Database identifies five putative maize PGK-encoding genes (https://corncyc-b73-v4.maizegdb.org/CORN/NEW-IMAGE?type=NIL&object=EC-2.7.2.3&redirect=T) including ZmPGK4 (Zm00001d043194) [45]. Our analysis showed that the predicted protein ZmPGK4 lacked key catalytic motifs (Figure 2). Comparing the different genomic assemblies, we confirmed that the missing domains were not due to incorrect annotation of exon or intron boundaries. Expression analysis indicates that *ZmPgk4* was not detectable in any of the analyzed tissues and stages. We also looked for *ZmPgk4* expression in three RNA-seq databases; [50,51,52]. From these, Sekhon et al. 2011 found any evidence of *ZmPgk4* expression [50]. We conclude that ZmPGK4 is not functional, due to lack of essential catalytic domains and limited transcript accumulation.

### 4.3. Nuclear ZmPGK5 Isozyme Is a Bacterial Protein Chimera with an Additional Domain

In addition to maize, several other plant species had PGK isozymes predicted to be nuclear-localized (*P. trichocarpa*, *R. communis*, *V. vinifera*, *S. bicolor,* and *O. sativa*) and having an additional methyltransferase domain (Appendix A). *Arabidopsis* lacked a nuclear isozyme, indicating that it may not be essential. The protein ZmPGK5 lacked four catalytic motifs but possessed a methyltransferase domain (Figure 2). It was steadily and weakly expressed in mostly all tissues (Figure 4). Therefore, it is unclear if ZmPGK5 plays a role in glycolysis or photosynthetic metabolism. When analyzing the PGK domain of ZmPGK5 separately from the methyltransferase domain, we found it to be more like a bacterial PGK than a plant cytoplasmic PGK (Appendix A). In addition, a BLAST search, using both domains individually, displayed high scores for cyanobacterial PGK and bacterial methyltransferase, which suggest ZmPGK5 may be a product of recombination after horizontal gene transfer. In *Pisum sativum*, it has been reported that a PGK protein can be part of the active DNA polymerase-α complex [60,61].

### 4.4. Molecular Elements Driving Differential ZmPgk Gene Expression

Gene expression is modulated by the presence and arrangement of cis-elements in promoter regions [62]. Notably, cis-elements involved in plastid, light, endosperm specific, and/or stress response gene regulation [35] were over-represented (Figure 5; Appendix A) in the promoter region of the ZmPGKs. Cis-elements involved in light-regulated gene expression were more abundant in plastid ZmPGK1, ZmPGK2, AtPGK1, and AtPGK2 (18 motifs per sequence) promoter sequences than those of cytosolic ZmPGK3 and AtPGK3 (12 motifs per sequence) and nuclear ZmPGK5 (7 motifs per sequence) genes (Figure 5; Appendix A). Surprisingly, cis-elements leading photosynthetic expression were found in all maize and *Arabidopsis* PGK promoter sequences independent of subcellular location (chloroplastic, cytosolic, and nuclear). This might explain why the cytosolic isozymes influence photosynthesis in *Arabidopsis* [56].

On average, promoter sequences contained a similar number of cis-elements: Plastid PGKs (22.5 motifs per sequence), cytosolic PGKs (21 motifs per sequence), and nuclear PGKs (23 motif per sequence) (Appendix A). Our results suggest that the expression of plastid *Pgk* genes in photosynthetic rather than non-photosynthetic tissues is due to ASF1MOTIFCAMV, IBOXCORE, TAAAGSTKST1, SORLIP1AT, CIACADIANLELHC, GT1CONSENSUS, SORLIP2AT, and INRNTPSADB elements (light-regulated) [35,63], together with the presence of chloroplast transit peptide and CACTFTPPCA1, GATABOX, -10PEHVPSBD, PRECONSCRHSP70A, and BOXIINTPATPB elements (plastid elements) [35,64]. *Pgk* gene expression may respond to adverse environmental conditions. ANAERO2CONSENSUS, MYCCONSENSUSAT, LTRECOREATCOR15, CAATBOX1 [35,65] elements indicated a role in maize and *Arabidopsis* acclimation.

In maize, several motifs were found to be involved in endosperm-specific genes during seed development. The maize RNA-seq database shows high expression of *ZmPgk3* in developing seeds from R1 to R4 stages (http://bar.utoronto.ca/efp_maize/cgi-bin/efpWeb.cgi?dataSource=Sekhon_et_al_Atlas). In contrast, plastid *ZmPgk* genes were almost not expressed in seeds. DOFCOREZM, EBOXBNNAPA, SEF4MOTIFGM7S, GCN4OSGLUB1, and ACGTCBOX elements [62] likely play an important role for *ZmPgk3* expression during maize seed formation. In the future, shuffling promotor elements and chip-sequencing assays might clarify the roles of each cis-element on *ZmPgk* gene expression.

### 4.5. Evolutionary Diversification of Plant PGKs

In cereal crops, sequence comparisons of the chloroplastic *PGK* genes have been used to determine phylogenetic relationships among the *Triticum* and *Aegilops* species of the wheat lineage, and to establish the timeline of wheat evolution and polyploidy [66]. Some authors have postulated that the photosynthetic PGKs emerged as the result of DNA recombination between eukaryotic and bacterial proteins [67,68], while others oppose this hypothesis [69]. Sequences comparison and phylogenetic analysis from the present study supports gene acquisition by duplication, whilst domain structure analysis supports protein relocation and mutagenesis in PGK plants. In Figure 8, we propose a model of evolution for *ZmPgk* genes. The chloroplastic and cytosolic ZmPGKs were more similar to each other (Figure 7; Table 1), which suggests that the chloroplast and cytosolic enzyme have not evolved directly from a prokaryotic progenitor, while the cytosolic form from animal, protozoa, and fungi have been inherited from a prokaryotic or eukaryotic progenitor.

Our analysis does not support the hypothesis that photosynthetic genes were directly acquired by horizontal transfer, albeit with minor kinetic modification, from a cyanobacterial ancestor [1], while glycolytic enzymes were inherited from a primitive eukaryotic host. Plants possess many more glycolytic isozymes than animals, fungi, and bacteria as a result of polyploidy and genomic duplication (Appendix A and Figure 6). The cyanobacterial PGKs are quite different from the photosynthetic plant PGKs (Figure 7). Gene duplication in maize may have helped to kinetically adjust the enzyme to regulatory particularities in pathways and maize tissues (Figure 6) that have opposite carbon fluxes and energy requirements (Figure 1A).

### 4.6. Distinct Roles of Plant PGKs across Species and Isozymes

In all sequenced plant genomes, PGK is encoded by a small gene family, but the functions of each isozyme remain to be further characterized [17]. Genomic surveys revealed three genes in *A. thaliana,* four genes in *S. bicolor* and *V. vinifera,* and five and seven genes in *P. trichocarpa* and *Oriza sativa* (Table 4), respectively. Plant species contained cytosolic and chloroplastic PGK isozymes (Table 4). Some species contained nuclear isozymes, while no species contained PGK in the mitochondria, nor in the vacuole, apoplast, or cell wall; thus, this central biochemical reaction might be absent from those compartments. In our phylogenetic analysis, glycolytic PGK isozymes were separated from photosynthetic PGK isozymes (Figure 7). For each type of PGK, specific branches emerged for monocot and dicot species (Figure 7), demonstrating that these gene duplications and sub-functionalization to different cellular compartments occurred prior to the divergence of these two lineages. Before, we showed that ZmPGK3 is crucial to produce ATP and recovers the energy investment of forming hexose phosphate into cell (Figure 2, Figure 3 and Figure 5). ZmPGK1 and ZmPGK2 are specific for plastid photosynthesis and they are predicted to consume ATP from the light reactions (Figure 2, Figure 3 and Figure 5), but ZmPGK1 play the main role in photosynthetic metabolism (Figure 6). ZmPGK4 looks to not be functional due to lack of essential catalytic domains and limited transcript accumulation (Figure 2 and Figure 3). ZmPGK5 is predicted to be in the nucleus with the addition of a protein domain (Figure 2 and Figure 3), which is not essential in plants, since *Arabidopsis* does not have a nuclear PGK (Table 4). In future studies, it would be interesting to perform in vitro and in planta characterization, in order to demonstrate the specific function of each ZmPGK isoforms.

## 5. Conclusions

We predict that ZmPGK3 is involved in glycolysis (ATP production) and that ZmPGK1 and ZmPGK2 are specific for plastid photosynthesis (ATP consumption). Duplication of plastid PGK isozymes has occurred in several plant species independently. In maize, the photosynthetic ZmPGK1 (mesophyll) was more highly expressed than ZmPGK2 (bundle sheath). In silico structural models reveal that the catalytically active ZmPGK isozymes have a high stereoselectivity to D-3PGA and adenine nucleotides with variable affinities (Km values) for ATP and 3PGA. Phylogenetic analysis of ZmPGKs with other PGKs from different species allowed glycolytic and photosynthetic isozymes to be distinguished. We propose a model based on several rounds of gene acquisition, duplication, protein relocation, mutagenesis, and gene deletion.

## Figures and Tables

**Figure 1 plants-09-01639-f001:**
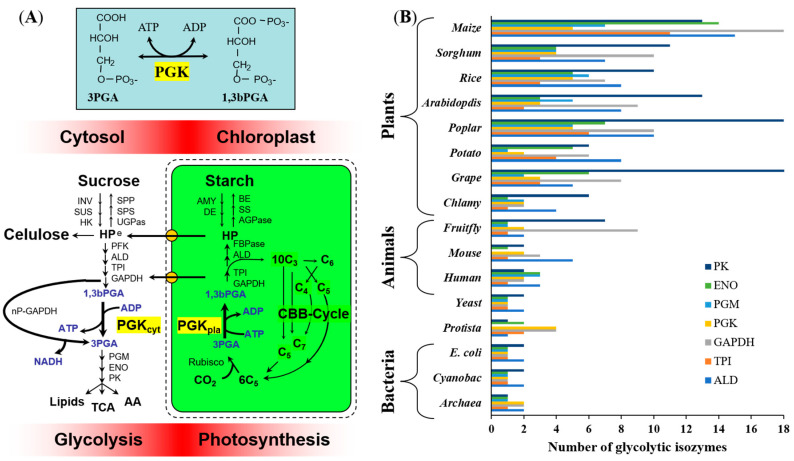
Isoenzymes of central carbon metabolism. (**A**) Schematic representation of the contribution of PGKs in the photosynthesis (consuming ATP), and glycolytic pathway (generating ATP). Invertase (INV), sucrose synthase (SUS), hexokinase (HK), Sucrose-6-P synthase (SPS), Sucrose-6-P phosphatase (SPP), UDP-glucose pyrophosphorylase (UGPase), Phosphofructokinase (PFK), Aldolase (ALD), triose-P isomerase (TPI), Glyceraldehyde dehydrogenase (GAPDH), non-phosphorylating GAPDH (nP-GAPDH), Phosphoglycerate mutase (PGM), Enolase (ENO), Pyruvate kinase (PK), Tricarboxilic acid cycle (TCA), amino acid synthesis (AA), Amylase (AMY), Debranching enzyme (DE), ADP-glucose pyrophosphorylase (AGPase), starch synthase (SS), Branching enzyme (BE), hexose-phosphate pool (HP), triose sugar phosphates (C3), treose sugar phosphates (C4), pentose sugar phosphates (C5), hexose sugar phosphates (C6), septose sugar phosphates (C7), Ribulose1,5,bP carboxylase oxigenase (Rubisco), Calvin-Benson-Bassham cycle (CBB). (**B**) Number of glycolytic isozymes (PK, ENO, PGM, PGK, GAPDH, TPI and ALD) from plants, animals and bacteria. The number of isozymes was obtained on January 2020 from https://www.plantcyc.org/ and https://biocyc.org/.

**Figure 2 plants-09-01639-f002:**
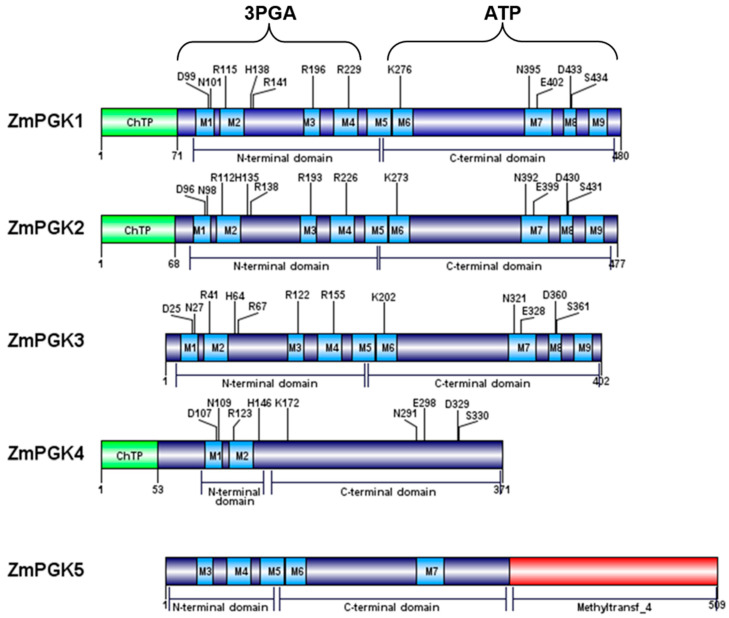
Domain structure of the *ZmPgk* gene family of maize. Blue boxes represent the nine conserved motifs (M1–M9) that are characteristic of PGKs from animals. Green boxes represent the chloroplast transit peptide (ChTP). The red box represents a different domain that corresponds to methyltransferase. Important residues involved in catalysis are represented by one letter code and their position inside the protein. ZmPGK4 lacks many of the conserved motifs and catalytic residues and therefore is likely to be a pseudogene. ZmPGK5 retains many conserved motifs but also has an additional domain.

**Figure 3 plants-09-01639-f003:**
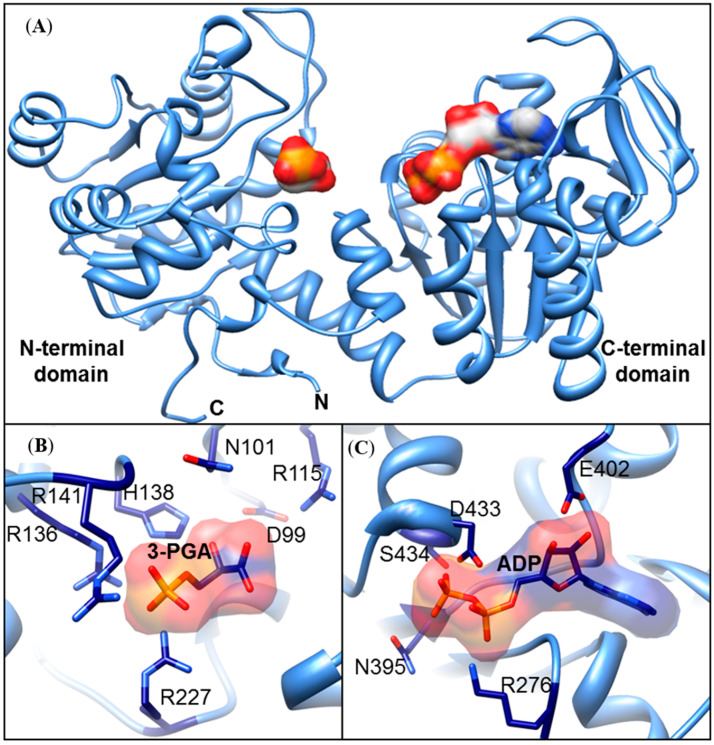
In silico structural model of maize ZmPGK1. (**A**) Interaction of ZmPGK1 with the substrates, 3PGA and ATP. The typical hinge (i.e., pacman) structure is observable between the N-terminal and C-terminal domains that close during catalysis. (**B**) The key residues that bind 3PGA are: D99, N101, R115, R136, H138, R141, and R227. (**C**) The key residues that bind ATP are: R276, N395, E402, D433, and S434.

**Figure 4 plants-09-01639-f004:**
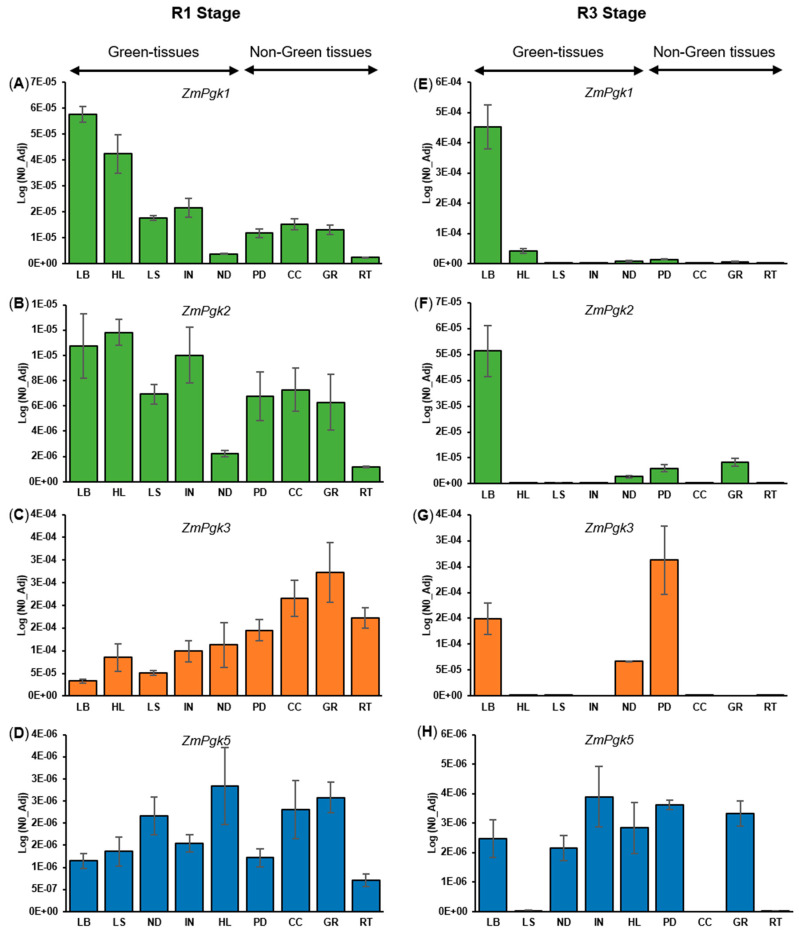
Gene expression analysis of *ZmPgks* in maize tissues of two reproductive development stages. Absolute quantification of plastid *ZmPgk1* and *ZmPgk2* (green bars), cytosolic *ZmPgk3* (orange bars), and nuclear (blue bars) genes were measured in R1 stage (**A**–**D**, respectively) and R3 stage (**E**–**H**, respectively). The tissues analyzed for each *ZmPgks* were leaf blade (LB), leaf sheath (LS), node (ND), internode (IN), husk leaf (HL), peduncle (PD), corncob (CC), grain (GR) and root (RT). Comparison of expression levels between *ZmPgks* and *Ef1α* (reference gene) in R1 and R3 stage are shown in Appendix A. *ZmPgk4* expression was not detectable in any of the analyzed tissues and stages. Bar represents the mean ± standard error (SE) with *n* = 3.

**Figure 5 plants-09-01639-f005:**
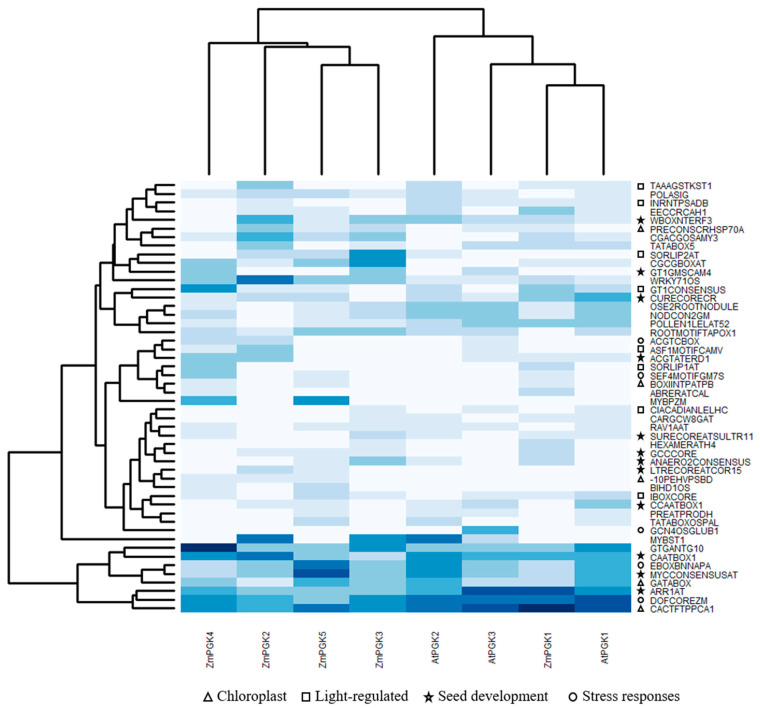
Heat map analysis, and cis-regulatory element annotation of ZmPGK and AtPGK promoter sequences. The most over-represented 50 motifs of both ZmPGK and AtPGK promoter sequences were used to build the heat map. The higher and lower number of motifs is represented by blue and white scale color, respectively (Appendix A). A sign to distinguish the elements involved in gene regulation of chloroplast, light-regulated, seed development and stress response genes was inserted.

**Figure 6 plants-09-01639-f006:**
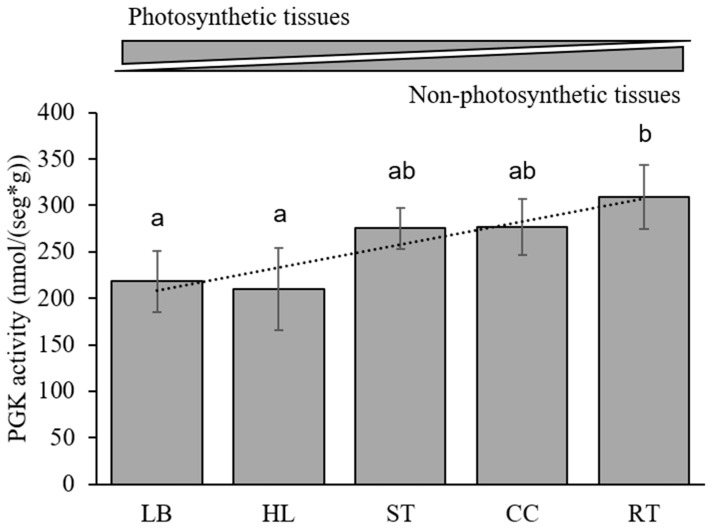
PGK activity in different tissues. PGK activity was measured in five maize tissues at R1 stage; Leaf blade = LB, Husk leaf = HL, Stem = ST (internode + node), Corncob = CC and Root = RT. Values = the average of three replicates (±SD), in units of activity per gram of fresh biomass.

**Figure 7 plants-09-01639-f007:**
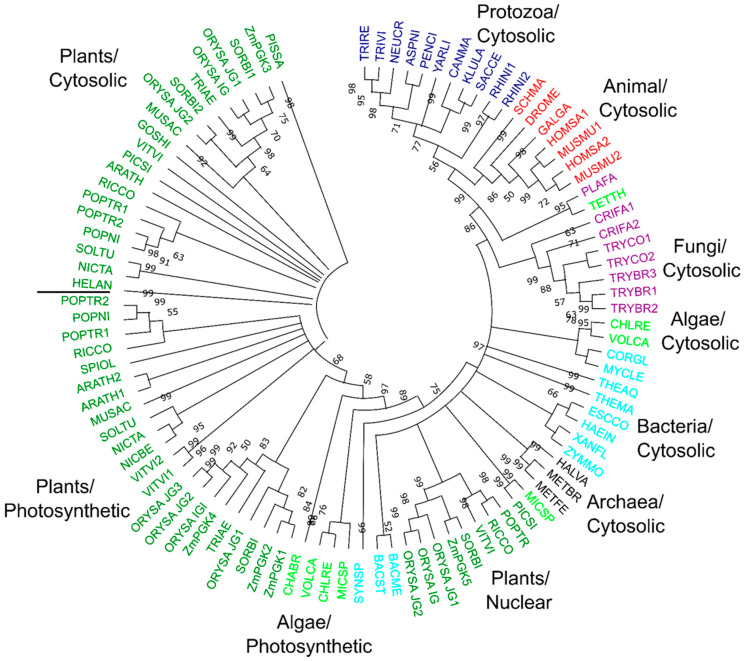
Maximum likelihood tree of 99 PGK protein sequences. The tree was constructed from a WAG model that used a gamma distribution for the variability of substitution rate across position. Numbers in nodes represent the bootstrap proportion for 1000 replicates of the same tree construction method. PGKs are colored according to the taxa as follow: Plants (green), Algae (yellow green), Fungi (purple), Protozoa (Navy blue), Animal (red), Bacteria (aquamarine), and Archaea (Black). Complete names of organisms, their accession numbers, and their lineages are listed in Appendix A. Phylogram is shown in Appendix A.

**Figure 8 plants-09-01639-f008:**
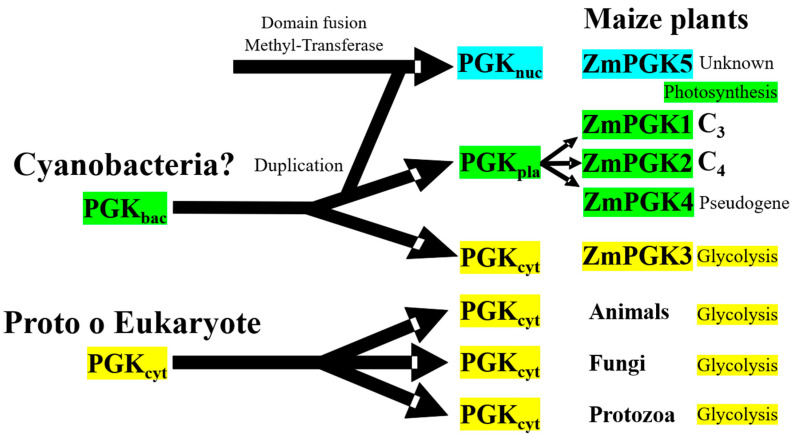
Proposed model of evolution for ZmPGK genes. Cyanobacterial ancestor is represented in green areas, whilst eukaryotic host is in yellow areas. Intracellular locations of PGKs; cyt = cytosolic, pla = plastid, and nuc = nuclear.

**Table 1 plants-09-01639-t001:** Maize PGK-encoding genes and protein identity percentage among them. Protein sequences were compared by pairwise alignments, using X-Clustal.

PGK-Encoding Genes		Protein Identity (% Amino Acid)			
		ZmPGK2	ZmPGK3	ZmPGK4	ZmPGK5
Zm00001d038579_T001.1	ZmPGK1	95.4	80.9	66.3	21.8
Zm00001d010672_T001.1	ZmPGK2	100	81.1	70.4	18.5
Zm00001d015376_T002.1	ZmPGK3		100	59.6	20.2
Zm00001d043194_T001.1	ZmPGK4			100	15.4
Zm00001d032867_T002.1	ZmPGK5				100

**Table 2 plants-09-01639-t002:** Properties of PGK isozymes in maize. The theoretical isoelectric point (pI) was calculated with ProtParam [27], using the EMBL pI tool from Lehninger (1979) and the Sequence Manipulation Suite [28]. Full-length protein sequences were analyzed for the presence of signal peptides, for subcellular localization, using TargetP v1.1^a^, Predotar v1.03^b^, and Wolf PSORT Prediction^c^. Different bioinformatic algorithms can make distinct predictions, given that some proteins can have multiple targeting to different compartments. Putative N-glycosylation sites were obtained from CBS prediction servers (http://www.cbs.dtu.dk/services/).

Protein Name	Molecular Weight kDa; Amino Acids (With, Without Signal Peptide)	Theoretical pI ProtParam EMBL SMS	Prediction of Location: Predotar ChloroP TargetP Wolf PSORT	Predicted N-Glycosylation Sites NetNGlyc 1.0 (Position Amino Acid)
ZmPGK1	49.8, 42.8; (480aa, 409aa)	6.29, 5.07 6.28, 4.88 6.69, 4.81	Plastid Chloroplast (71aa) Chloroplast (71aa) Chloroplast	109 NITD
ZmPGK2	49.6, 42.9; (477aa, 409aa)	6.99, 5.13 7.36, 4.93 7.62, 4.88	Plastid Chloroplast (68aa) Chloroplast (68aa) Chloroplast	106 NITD
ZmPGK3	42.4; (402aa)	5.65 5.52 5.58	None None None Cytoplasm	No sites predicted
ZmPGK4	38.3, 32.9; (371aa, 318aa)	7.02, 5.80 7.26, 5.73 7.55, 6.05	Plastid Chloroplast (53aa) Chloroplast (53aa) Chloroplast	22 NSTG 117 NITD
ZmPGK5	56.4; (509aa)	6.19 6.15 6.50	None None None Nucleus	39 NFTG 63 NDSF 202 NSTG 389 NATS

**Table 3 plants-09-01639-t003:** Estimation of substrate affinity to 3PGA and/or ATP for maize tissues.

Tissue	Km PGA (mM)	Km ATP (mM)
Leaf	0.84 ± 0.11	0.06 ± 0.01
Husk	0.47 ± 0.06	0.24 ± 0.07
Stem	0.59 ± 0.10	0.08 ± 0.02
Cob	0.45 ± 0.06	0.24 ± 0.07
Root	0.50 ± 0.07	0.08 ± 0.02

**Table 4 plants-09-01639-t004:** Orthologous PGK proteins of different plant species. The accession numbers were obtained from TAIR (*Arabidopsis thaliana*), MaizeSequence (*Zea mays*), GenBank, and Ensemble (*Vitis vinifera, Sorghum bicolor, Populus trichocarpa*, and *Oriza sativa* Japonica Group).

Localized	*Monocots*			*Dicots*		
	*Z. mays* (C4)	*S. bicolor* (C4)	*O. sativa* (C3)	*A. thaliana* (C3)	*V. vinifera* (C3)	*P. trichocarpa* (C3)
Plastid/Photosynthetic	ZmPGK1 Zm00001d038579 ZmPGK2 Zm00001d010672 ZmPGK4 Zm00001d043194	SbPGKpla1 SORBI_3009G183700	OsPGKpla1 AAT07576.1 OsPGKpla2 EEE55536.1 OsPGKpla3 NP_001172606.1	AtPGK1 AT3G12780 AtPGK2 AT1G56190	VvPGKpla1 XP_003634814.1 VvPGKpla2 XP_002263796.1	PtPGKpla1 POPTR_010G171500v3 PtPGKpla2 POPTR_008G084500v3
Cytoplasmic/Glycolytic	ZmPGK3 Zm00001d015376	SbPGKcyt1 SORBI_3004G055200 SbPGKcyt2 SORBI_3010G221800	OsPGKcyt1 ABL74575.1 OsPGK7cyt2 NP_001058317.1	AtPGK3 AT1G79550	VvPGKcyt1 XP_002263950.1	PtPGKcyt1 POPTR_008G084400v3 PtPGKcyt2 POPTR_010G171600v3
Nuclear	ZmPGK5 Zm00001d032867	SbPGKnuc SORBI_3001G236800	OsPGK5nuc1 ABB47707.1 OsPGK6nuc2 EEE51029.1	-	VvPGKnuc CBI32769.3	PtPGKnuc POPTR_016G091800v3

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
