# Peer review of "The Phosphoglycerate Kinase (PGK) Gene Family of Maize (Zea mays var. B73)"

_plants, 2020, doi:10.3390/plants9121639_

Round 1
Reviewer 1 Report
In this paper, the authors described some important aspects of phosphoglycerate kinase gene family in maize. However, this paper is not clearly written which can easily understandable for the readers.
e.g. Title is very broad and not complete. The authors should make it clear which aspect of PGK is discussed in this paper.
Discussion should be improved. The implication of PGK is not clearly discussed here.
The paper is fine but I was suggesting changing the title and proofread for linguistic and formatting errors. Also improving the Introduction and Discussion citing vital and please cite recent references. The background and the impact of the research should be clearly mentioned in the introduction. Putting statistical lettering on the bars (In figures) was suggested.Author Response
Please see the attachment

Reviewer 2 Report
Massange-Sánchez et al in their manuscript provide functional, structural and phylogenetic analysis of the phosphoglycerate kinase (PGKs) in maize. Authors identify five homologous PGK genes in the reference genome of Zea mays var. B73. Authors perform thorough bioinformatics analysis for these proteins: predict their cellular localization, domains and active sites in sequence and reconstructed 3D models, identify cis-regulatory elements and determine the expression patterns of these genes from the gene expression databases. They experimentally estimate the expression in various organs of plants at two growth stages and estimated the PGK activity. Authors propose the model of PGK evolution in plants based on the obtained data. Overall, the manuscript represent integral view on the structure, function and evolution of these genes. The work suggests that two of Z. mays PGKs (ZmPGK1 and ZmPGK2) are specific for plastid photosynthesis, ZmPGK3 is involved in glycolysis, ZmPGK4 is a pseudogene and ZmPGK5 has functions unrelated to glycolysis or photosynthetic metabolism.
I have several questions/recommendations to authors.
(A) Phylogenetic analysis:
- I suggest using additional method for phylogeny reconstruction, for example, IQ-TREE, because it implements broad range of substitution models and allows selecting the best one.
- I also recommend represent the phylogenetic tree (Fig 6) as a phylogram (in a supplementary figure) in addition to cladogram, since branch lengths could be informative.
- Another suggestion is to make colors of the leaf nodes (Fig 6) according to taxa, and name clusters according their common taxon/function. For example, from the Fig 6 it could be seen that some algae chloroplastic PGKs (CHLRE cyt; VOLCA Cyt) colored in red (as bacterial nodes, but they are not bacterial PGKs). Interestingly, they both closer to bacterial PGKs, than for plant chloroplastic PGKs. How to explain this?
- It would be also interesting to include in the phylogenetic analysis sequences from multicellular green algae (charophytes), which are regarded as the ancestors of land plants.
(B) Additional ZmPGK5 methyltransferase domain:
- “the PGK domain of ZmPGK5 … more like a bacterial PGK than a plant cytoplasmic PGK”, but did not present a data supporting this hypothesis. I suggest either remove this phrase (and change the subsection title) or provide the data supporting this hypothesis (in the supplementary files). I also wonder whether the methyltransferase domain presented in other PGKs (plant, plants:Nuclear or non-plant)? It is implied in Fig 7.
Minor concerns.
- I recommend to introduce additional column in Supplementary table S1 describing the taxon from Fig. 6, after the sequence length and before the lineage (which is difficult to read).
- Please, provide color bar description for Fig 4. What do they mean?
- Line 301: delete ‘(they)’.
- Fig 7: What do you mean by ‘Ancient plants’? Is it the most recent green plant ancestor? What do you mean by ->X and -> X replacement?
Reviewer 3 Report
The manuscript titled “The phosphoglycerate kinase (PGK) gene family of maize (Zea mays var. B73)” by Massange-Sánchez et al. identified putative phosphoglycerate kinase genes in Zea mays var. B73 and characterized them using in silico, biochemical, and phylogenetic studies. The finding is interesting and contributes to the study field. However, some critical information is missing or is not clearly presented-- these should be supplied to validate the finding. In addition, I feel that the authors claim more than what the presented data and evidence could, requiring rewriting such parts. In particular, the interpretation and claims regarding the evolution of ZmPGKs require more sequencing analyses as direct evidence. Without it, any currently made claims should be toned-down and presented as conjectures or hypotheses. In addition, I ask authors to address below:
How are the putative ZmPGKs identified in this study related to the originally reported PGK in Zea mays (NCBI reference sequence: NP_001167912.1)? Is one of the ZmPGKs the same one as the NCBI registered one? PUBMED ID: 19965430. This is critical to cite the paper and compare these proteins.
Figure S1 (left panel) is important information to understand the manuscript and so it should be placed as Figure 1. Please make it sure that the figure is authentic (not a copy from any published sources).
Split the left and right panels of Figure S1 into Figure 1A and 1B.
“Structural model” should be substituted to in silico structural model throughout the manuscript and figures.
There is no description of development stages named of R1–R4 in the main text.
ZmPgk gene expression analysis using the samples from the R3 stage appears not valid for 2, 3, 4, 7, 9 (Figure S4 and Figure 3) because signals of a reference gene seem barely detectable for these tissues in the current presentation.
What do authors mean for “absolute quantification”? Are these relative values for a reference gene, EF1a in each tissue? The graph should be revised to show relative fold expression (as Y axis) to reference gene in all qRT-PCR data in figures.
Lines 307–310: direct evidence for the claim (functions of the putative ZmPGKs) was not provided in the study and so it is yet a conjecture, requiring further studies. Please rewrite it.
Line 313, a reference is missing.
Lines 339–340: ZmPGK5 doesn’t seem to be expressed in all tissues as claimed by authors (Figure 3) and ubiquitous expression doesn’t mean that the protein doesn’t have an essential role. The bacterial gene transfer could be a hypothesis but the authors did not provide with any solid evidence to suggest it. This paragraph should be removed or rewritten to be true to the facts and to separate the facts from a hypothesis/conjecture.
I cannot locate the information of the sources of proteins (e.g. GAPDH) and reagents (ATP, PGA, NADH) used for PGK enzyme activity measurement. The authors should also include the raw data including the controls in the supplementary information.
Line 358: the numbers of motifs are not the same.
Lines 381–382: Where are the sequence comparisons (e.g. between cyanobacterial PGKs and ZmPGKs) suggest the gene acquisition by duplication, protein relocation, mutagenesis, and gene deletion? This claim requires more evidence to be presented.
Show the enzyme kinetics raw data and Michaelis-Menten graphs to draw KM values along in Figure 5.
Lines 384–389: Please rewrite to be clearer. Also, what are the protein sizes to do with the origin? Do authors, in Figure 7, point that ZmPGKs are from cyanobacteria? The evolution part is overall confusing and less convincing.
Lines 155–162: How homologous are the putative ZmPGKs to E. coli and human PGKs in percentage?
Line 401: A reference is missing or not in a consistent format.
Line 406: E.C. 2.7.2.3 ïƒ What are those? All of these require the sources throughout the manuscript.
Lines 416–417: Do authors mean that Arabidopsis is viable without PGK5? But there is no PGK5 in Arabidopsis.
Round 2
Reviewer 2 Report
Authors answered all the questions raised in the review.
I only have minor suggestion related the Fig S6:
Please, use unrooted tree representation in the Figure and show bootstrap values near the branches.
Reviewer 3 Report
For purchased proteins including NADH, GAPDH and BSA and ATP, the catalog numbers should be provided in the materials and methods along with their vendor information.
For ZmPGK4, indicate the NCBI protein ID number in line 190, along with the reference 45.
Line 351-352 is missing a reference or not in a consistent format. There are few studies about ZmPGK, I believe. Rewrite “only Sekhon et al. 2011 found any evidence of ZmPgk4 expression to be less exaggerated. In this paragraph, maybe in line 347, add the reference 45.
In the response, the authors argue “Regards ZmPGK5 doesn’t have an essential role in glycolysis or photosynthesis is because its predicted to be located at nucleus, unlike to ZmPGK1 and ZmPGK2 predicted to be target at chloroplast or ZmPGK3 predicted to stay at cytoplasm.” In the text, “these data suggest that ZmPGK5 does not have an essential role in glycolysis or photosynthetic metabolism” should be backed up by clearer reasons. Please articulate them.
In lines 408-406, ZmPGKs are ~450 or 470 aa??
If DNA duplication occurs in evolution, the protein sizes could be larger. Not sure how the authors could exclude the possibility from bacterial origins by this. I think that the evolutional analyses and interpretation in 4.5 should be more rigorous not to be wrong or too much stretched.
Are the below the homology of DNA or protein??
Point 16: Lines 155–162: How homologous are the putative ZmPGKs to E. coli and human PGKs in percentage?
Response 16: Here the information required.
|
E. coli K12 |
ZmPGK5 |
H. sapiens |
ZmPGK3 |
ZmPGK4 |
ZmPGK2 |
ZmPGK1 |
|
|
E. coli K12 |
100 |
14.24 |
15.5 |
17.73 |
20.15 |
19.48 |
19.2 |
|
ZmPGK5 |
14.24 |
100 |
24.92 |
24.37 |
24.41 |
26.17 |
26.48 |
|
H. sapiens |
15.5 |
24.92 |
100 |
49.62 |
45.99 |
48.87 |
49.12 |
|
ZmPGK3 |
17.73 |
24.37 |
49.62 |
100 |
76.12 |
80.55 |
80.3 |
|
ZmPGK4 |
20.15 |
24.41 |
45.99 |
76.12 |
100 |
84.59 |
84.25 |
|
ZmPGK2 |
19.48 |
26.17 |
48.87 |
80.55 |
84.59 |
100 |
95.35 |
|
ZmPGK1 |
19.2 |
26.48 |
49.12 |
80.3 |
84.25 |
95.35 |
100 |
Provide the clearer information and homology both in DNA and in protein levels.
